# Structure of MlaFB uncovers novel mechanisms of ABC transporter regulation

Ljuvica R Kolich[1†], Ya-Ting Chang[1†], Nicolas Coudray[1,2†], Sabrina I Giacometti[1], Mark R MacRae[1], Georgia L Isom[1], Evelyn M Teran[1], Gira Bhabha[1], Damian C Ekiert[1,3]*

[1]Department of Cell Biology, New York University School of Medicine, New York, United States; [2]Applied Bioinformatics Laboratory, New York University School of Medicine, New York, United States; [3]Department of Microbiology, New York University School of Medicine, New York, United States

**Abstract** ABC transporters facilitate the movement of diverse molecules across cellular membranes, but how their activity is regulated post-translationally is not well understood. Here we report the crystal structure of MlaFB from *E. coli*, the cytoplasmic portion of the larger MlaFEDB ABC transporter complex, which drives phospholipid trafficking across the bacterial envelope to maintain outer membrane integrity. MlaB, a STAS domain protein, binds the ABC nucleotide binding domain, MlaF, and is required for its stability. Our structure also implicates a unique C-terminal tail of MlaF in self-dimerization. Both the C-terminal tail of MlaF and the interaction with MlaB are required for the proper assembly of the MlaFEDB complex and its function in cells. This work leads to a new model for how an important bacterial lipid transporter may be regulated by small proteins, and raises the possibility that similar regulatory mechanisms may exist more broadly across the ABC transporter family.

**\*For correspondence:**
damian.ekiert@EKIERTLAB.ORG

[†]These authors contributed equally to this work

**Competing interests:** The authors declare that no competing interests exist.

## Introduction

ABC transporters catalyze the import or export of a wide range of molecules across cellular membranes, including ions (*Riordan et al., 1989*), drugs (*Juliano and Ling, 1976*), antibiotics (*Fitzpatrick et al., 2017*), sugars (*Ehrmann et al., 1998*), lipids (*Polissi and Georgopoulos, 1996*; *Brooks-Wilson et al., 1999*; *Rust et al., 1999*; *Bodzioch et al., 1999*; *Berge et al., 2000*), and even large proteins (*Kanonenberg et al., 2018*). Named for their conserved A̲TP B̲inding C̲assette (ABC) nucleotide binding domains (NBD), ABC transporters use energy from ATP hydrolysis by the NBDs to drive conformational changes in their associated transmembrane domains (TMD) in order to translocate substrates across a lipid bilayer. Levels of transporter activity can be regulated both at the level of transcription/translation as well as post-translationally (*Du et al., 2018*), allowing cells to fine tune the import and export of specific metabolites in response to changing environmental conditions. Post-translational regulatory mechanisms are particularly powerful, as they allow the cell to respond rapidly by activating or inactivating existing transporter complexes, in contrast to the relatively slow process of transcribing, translating, and assembling new transporters. ABC transporters are frequently regulated post-translationally through direct modification of transporter subunits, such as by phosphorylation or ubiquitination (*Crawford et al., 2018*). However, a small but growing number of regulatory proteins have been shown to stimulate or inhibit ABC transporter activity, such as MetNI (*Kadaba et al., 2008*), ModBC (*Gerber et al., 2008*), MalFGK (*Deutscher et al., 2006*; *Chen et al., 2013*), and CFTR (*Walter et al., 2019*; *Ko et al., 2004*). These interaction-based mechanisms of transporter regulation may be quite common but underrepresented in the literature, as

identification of protein-protein interactions involving membrane proteins is challenging, and frequently screens for protein-protein interactions are biased toward soluble proteins (e.g., yeast 2-hybrid).

An ABC transporter is a critical component of the Mla pathway, which is a phospholipid transport system that is important for outer membrane (OM) integrity in many Gram-negative species, and has been implicated in the <u>M</u>aintenance of outer membrane <u>L</u>ipid <u>A</u>symmetry (*Malinverni and Silhavy, 2009*; *Thong et al., 2016*; *Kamischke et al., 2019*; *Chong et al., 2015*; *Sutterlin et al., 2016*; *Ekiert et al., 2017*). The Mla system from *E. coli* consists of three main parts: 1) an inner membrane (IM) ABC transporter complex, MlaFEDB (*Thong et al., 2016*; *Ekiert et al., 2017*; *Isom et al., 2017*); 2) an OM complex, MlaA-OmpC/F (*Chong et al., 2015*; *Abellón-Ruiz et al., 2017*); and 3) a periplasmic shuttle protein, MlaC (*Ekiert et al., 2017*), which ferries lipids between the IM and OM complexes (*Figure 1A*). Mla was originally proposed to drive the import of mislocalized phospholipids from the outer leaflet of the OM toward the IM, thereby maintaining the asymmetry of the OM (*Malinverni and Silhavy, 2009*). However, more recent work has suggested that Mla may instead drive the export of newly synthesized phospholipids to the OM as the cell grows (*Kamischke et al., 2019*; *Hughes et al., 2019*), and the direction of lipid transport mediated by Mla remains to be firmly established.

Biochemical analysis and low-resolution structures of the MlaFEDB complex from *E. coli* (*Thong et al., 2016*; *Ekiert et al., 2017*) and *A. baumanii* (*Kamischke et al., 2019*) suggest that a total of 12 polypeptide chains associate to form the complex, with an overall stoichiometry of Mla-$F_2E_2D_6B_2$. Two copies of MlaF, an ABC ATPase, and two copies of MlaE, a multipass transmembrane protein, form an $MlaF_2E_2$ subcomplex that represents the conserved ABC transporter core. Six copies of the MlaD subunit assemble into a hexameric ring of MCE domains, forming an unusually large periplasmic/extracytoplasmic domain that has been proposed to serve as a tunnel for lipid transport between $MlaF_2E_2$ and periplasmic shuttle protein, MlaC (*Ekiert et al., 2017*). In contrast, the function of the MlaB subunits are poorly defined. MlaB is stably associated with the cytoplasmic part of the transporter complex (*Thong et al., 2016*): one MlaB subunit binds to the side of each of the two MlaF ATPase domains (*Kamischke et al., 2019*; *Ekiert et al., 2017*). However, given the low resolution of the MlaFEDB structures determined to date, MlaB could not be built de novo, and homology models of MlaB docked in the maps could not be positioned reliably, providing minimal insights into the interaction between MlaF and MlaB. We and others have hypothesized that MlaB may regulate the MlaFEDB transporter complex (*Thong et al., 2016*; *Ekiert et al., 2017*), perhaps through its interaction with the MlaF ATPase. Here we report structures of the MlaFB complex, and present a model for how these two proteins may work together.

## Results

### Overview of the MlaFB crystal structure

MlaF and MlaB are both required for the proper functioning of the Mla transport system in *E. coli* (*Malinverni and Silhavy, 2009*; *Thong et al., 2016*; *Ekiert et al., 2017*). Consistent with these previous reports, an in-frame deletion of *mlaF* from the *E. coli* chromosome compromises outer membrane integrity, leading to poor growth of the mutant relative to the wild type strain on LB agar in the presence of OM perturbing agents such as SDS and EDTA, which can be complemented by providing a wild type copy of *mlaF* on a plasmid (*Figure 1B*; *Supplementary file 1* and *Supplementary file 2*). Similarly, a strain with an in-frame disruption of *mlaB* also grows poorly in the presence of SDS and EDTA (*Figure 1B*), though the *mlaB* phenotype is slightly milder. We observed that the optimal amount of SDS and EDTA for detecting *mla* phenotypes was strongly dependent on the source of LB agar (see methods, *Figure 1—figure supplement 1*), which may explain why the reported conditions vary between studies (*Malinverni and Silhavy, 2009*; *Thong et al., 2016*; *Ekiert et al., 2017*).

To better understand how the MlaB and MlaF subunits interact, and how this interaction may impact function of the MlaFEDB transporter, we determined crystal structures of MlaF in complex with MlaB. To produce the complex for crystallization, MlaF and MlaB were co-expressed from a single transcript, and purified by $Ni^{2+}$ affinity chromatography via an N-terminal His tag on MlaB (*Figure 1C*). The MlaF and MlaB subunits co-eluted when further purified by size exclusion

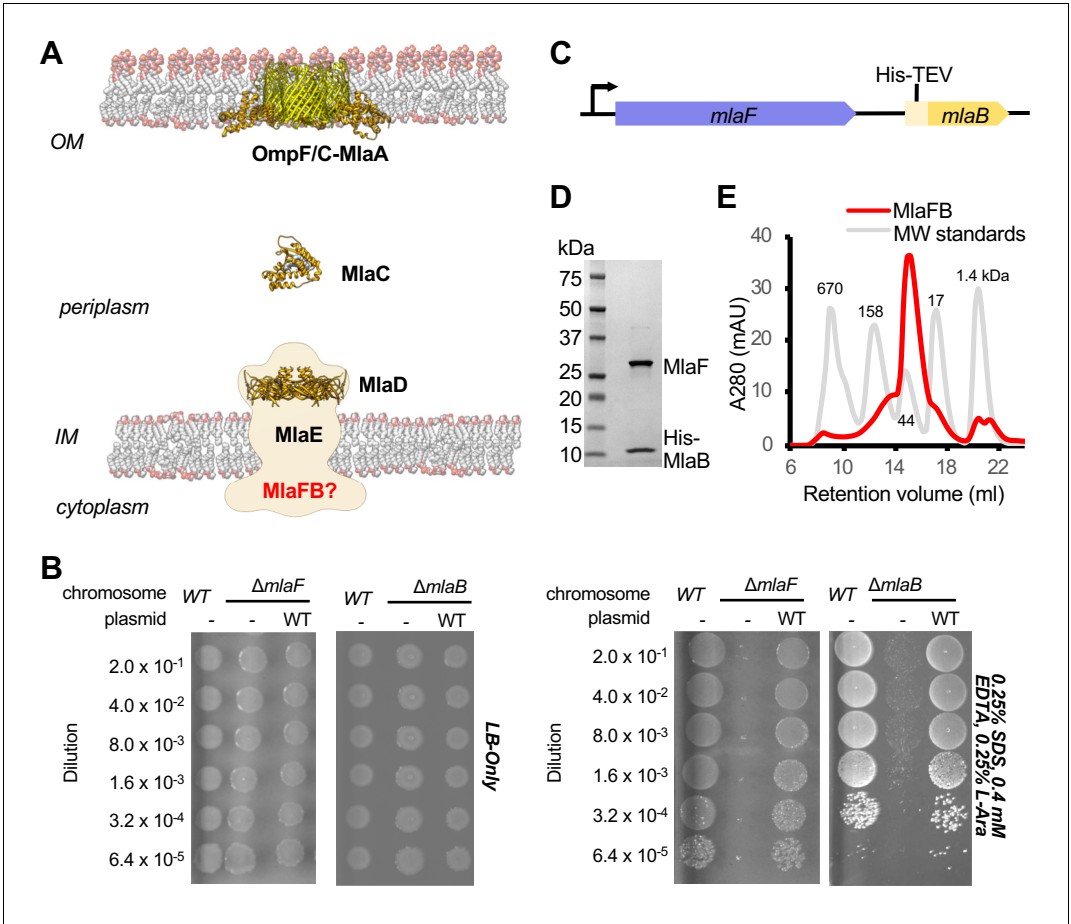

**Figure 1.** Characterization of MlaFB complex. (**A**) Schematic of the Mla pathway including the soluble periplasmic protein MlaC (PDB 5UWA), outer membrane complex, OmpF/C-MlaA (PDB 5NUP) and integral inner membrane complex, MlaFEDB (PDB 5UW2, EMDB-8610). (**B**) Cellular assay for the function of MlaF and MlaB. 10-fold serial dilutions of the indicated cultures spotted on LB only (top) or LB plates containing SDS and EDTA at the concentrations indicated (bottom) and incubated overnight. The *mlaF* and *mlaB* single knockouts grow poorly in the presence of SDS+EDTA, but can be rescued by the expression of WT mlaF or mlaB constructs, respectively. Results depicted are representative of at least three biological replicates. (**C**) Schematic of the MlaFB operon. (**D**) SDS PAGE and (**E**) size exclusion chromatogram of purified MlaFB complex.

The online version of this article includes the following figure supplement(s) for figure 1:

**Figure supplement 1.** Optimization of conditions for growth analysis on different agar/media.

**Figure supplement 2.** Docking of ADP-bound MlaF$_2$B$_2$ complex into previous low resolution EM density map of *E. coli* MlaFEDB (EMD-8610).

---

chromatography with an apparent molecular weight of approximately 40 kDa, consistent with a 1:1 MlaFB heterodimeric complex (expected 41.4 kDa; *Figure 1D,E*). Although a 1:1 complex in solution, MlaFB crystallized as both a 1:1 complex (MlaF$_1$B$_1$) and a 2:2 complex (MlaF$_2$B$_2$), and we solved MlaFB crystal structures of both MlaF$_1$B$_1$ (2.60 Å, no nucleotide bound, PDB code: 6XGZ) and MlaF$_2$B$_2$ (2.90 Å, ADP bound, PDB code: 6XGY) (*Supplementary file 3*). Here, we focus on the MlaF$_2$B$_2$ complex, which is fully assembled as expected for an ABC transporter ATPase subunit. In contrast, the MlaF$_1$B$_1$ structure represents an inactive 'half-transporter', but will be discussed where it provides additional insights into unique features of the MlaF-MlaB interaction. These data enabled the building of a high quality model of the MlaF$_2$B$_2$ subcomplex that is globally consistent with the overall domain locations inferred from low resolution EM studies of *E. coli* MlaFEDB (*Figure 1—figure supplement 2*), where the relative orientation of the MlaB and MlaF subunits could not be assigned (*Kamischke et al., 2019*; *Ekiert et al., 2017*). The final model for MlaF$_2$B$_2$ consists of MlaF residues 5–267 (of 269 total) and MlaB residues 2–97 (of 97 total), as well as an ordered portion of

the N-terminal purification tag [Prior to publication, coordinates are available for download from our website: URL].

At the core of the MlaF$_2$B$_2$ complex is an MlaF homodimer, assembled in a 'head-to-tail' arrangement similar to the NBDs of other ABC transporters in the dimeric state (*Figure 2A*). Two ADP molecules are bound at the dimer interface (*Figure 2*, A and C), along with two Mg ions, representing the post-powerstroke state following ATP hydrolysis and phosphate release. The MlaB subunits bind near the periphery of the MlaF dimer and make no direct contacts with each other. This results in a

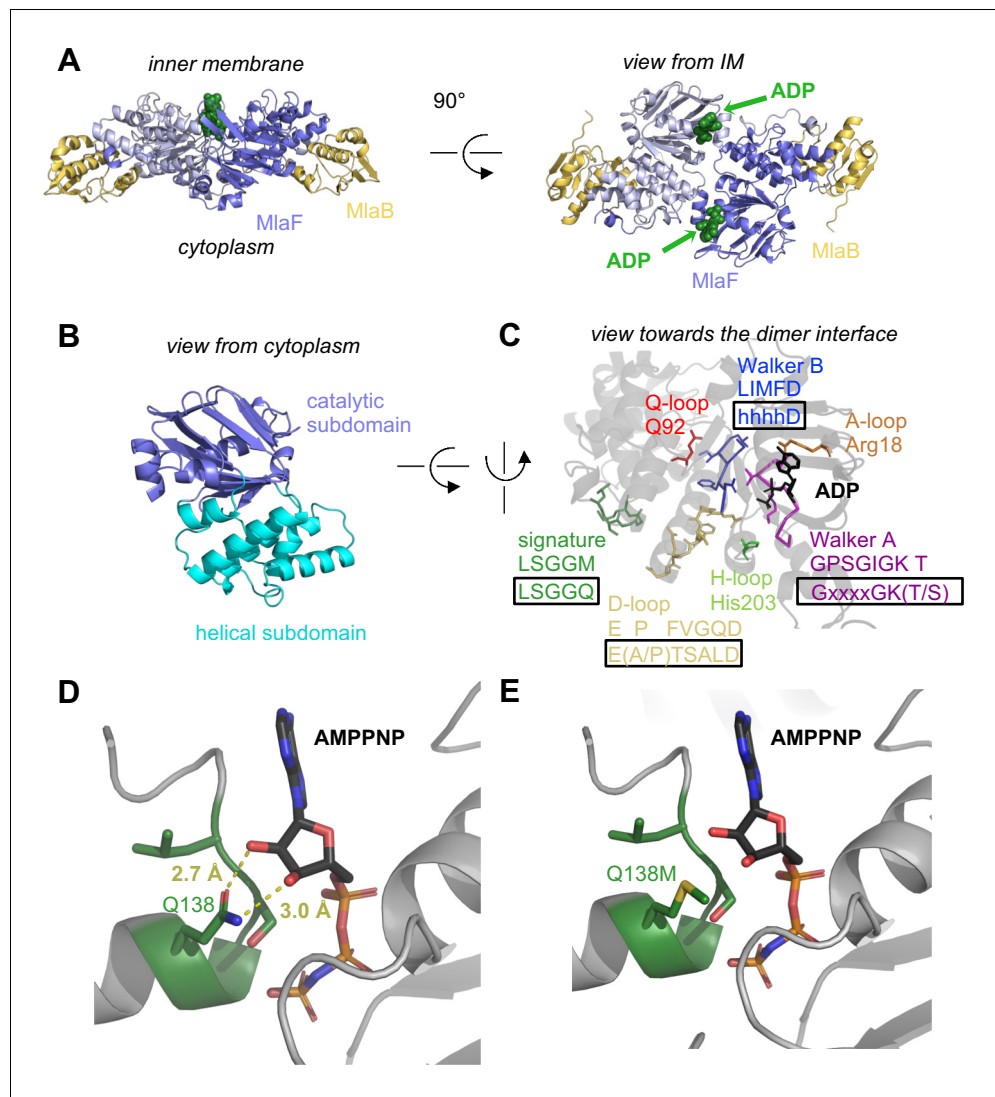

**Figure 2.** Crystal structure of ADP-bound heterodimeric MlaF$_2$B$_2$. (A) Side and top view of the complex, formed by a dimer of MlaFB heterodimers; MlaF (purple) and MlaB (yellow) are shown in cartoon representation and ADP (green) is shown as spheres. (B) The two subdomains of MlaF. (C) Functionally important motifs in ABC domains highlighted for one protomer of MlaF with sequence shown directly below the motif name, and the consensus sequences displayed in rectangles. (D) Gln side chain of LSGGQ/signature motif, which is within hydrogen bonding distance of the 2' and 3' hydroxyls of ATP ribose moiety in many ABC transporter structures (depicted here, PDB: 3RLF [*Oldham and Chen, 2011*]). (E) Q138M mutation modeled in 3RLF, which shows that a Met at this position would be unable to make the same interactions with ATP.

The online version of this article includes the following figure supplement(s) for figure 2:

**Figure supplement 1.** Position of Q-loop and subdomain conformations in different structures of MlaF.
**Figure supplement 2.** Sequence alignment of Mla, TGD-like MCE systems, o-antigen and teichoic acid transporter.

complex that is distinctly elongated in one direction, with the two MlaB subunits forming a sandwich with the MlaF dimer in the middle.

## MlaF is an ABC ATPase

MlaF exhibits a canonical ABC ATPase fold, consisting of two subdomains: a mixed α/β catalytic core shared among P-loop ATPases and a helical subdomain characteristic of the ABC family (*ter Beek et al., 2014*; *Figure 2*, A and B). MlaF is approximately 30% identical to the most closely related ABC ATPase structures from the PDB (1OXS [*Verdon et al., 2003*], 3DHW [*Kadaba et al., 2008*], and 1Z47 [*Scheffel et al., 2005*]), and our crystal structures of MlaF can be superimposed on these structures with an RMSD of ~1.5–2.5 Å, consistent with a similar transport function. The key motifs required for ATP hydrolysis and coupling to transport (*ter Beek et al., 2014*) are well conserved in MlaF and clearly resolved in our electron density map (*Figure 2C*). Our MlaF structure adopts an open conformation capable of nucleotide exchange, with 13.9 Å separating the P-loop/Walker A motif in one subunit from the ABC signature motif in the opposing subunit (between Cα of Gly46 and Gly147). The catalytic and helical subdomains of ABC-type ATPases are known to rotate relative to one another in response to nucleotide binding (*Karpowich et al., 2001*; *Smith et al., 2002*; *Orelle et al., 2010*) and upon interaction with the coupling helices of the TMDs. Indeed, when our $MlaF_1B_1$ and $MlaF_2B_2$ structures are superimposed on the catalytic subdomain, we observe a rotation of the helical subdomain by ~10 degrees (*Figure 2—figure supplement 1A*). Perhaps as a consequence of this subdomain rotation, the Q-loop is displaced from its usual location near the site of catalysis (*Figure 2—figure supplement 1B*), consistent with its role in linking the site of ATP hydrolysis in the catalytic subdomain to the helical subdomain.

There is a noteworthy difference in the final residue of MlaF's signature motif, which differs from the consensus sequence (MlaF: 145-LSGGM-149; consensus: LSGGQ). In structures of ABC transporters captured an ATP bound state, such as the *E. coli* maltose transporter (*Oldham and Chen, 2011*), the signature motif of each NBD clamps the bound nucleotide against the core catalytic motifs provided by the adjacent subunit, and the consensus glutamine residue may hydrogen bond to the 2' and 3' hydroxyls of the ATP ribose (*Figure 2D*). As methionine at this position in MlaF could not hydrogen bond in the same way as glutamine (*Figure 2E*), this substitution may impair the ability of MlaF to discriminate between ATP and deoxyATP, or decrease its affinity for NTPs. However, the functional significance of the Gln149Met substitution is not clear from our structure and we cannot rule out the possibility that other nearby residues may substitute for the conserved glutamine. While methionine is very rare at this position in structurally characterized ABC transporters (5 of 191 sequences analyzed, whereas Gln/Glu is present in the remaining 183), it is well-conserved in the signature motifs of NBD sequences from Mla- and TGD-like MCE systems (*Figure 2—figure supplement 2*). Methionine is also found at this position in the O-antigen (*Bi et al., 2018*; *Caffalette et al., 2019*) and teichoic acid (*Chen et al., 2020*) transporters, suggesting they may have functional similarity to the Mla system.

## MlaF forms a domain swapped dimer via a unique C-terminal extension

Intriguingly, each MlaF subunit possesses a unique ~25 aa C-terminal extension (CTE) that wraps around the neighboring MlaF subunit, forming a domain swapped reciprocal 'handshake', and interacting with the adjacent MlaB as well (*Figure 3A*). Much of the C-terminal segment adopts an extended 'random coil' structure, before terminating in a short helix that binds to the neighboring MlaFB module, where it inserts three hydrophobic residues into a groove at the MlaF-MlaB interface (Tyr261, Leu265, Leu266; *Figure 3A*). No crystal contacts are mediated by the MlaF CTE, suggesting that its interaction with neighboring subunits is not merely due to crystal packing. Moreover, in the $MlaF_1B_1$ structure, which crystallized in a different space group ($P2_12_12_1$) from that of $MlaF_2B_2$ ($P3_221$) with completely different crystal packing, the MlaF CTE is ordered and the terminal helix is bound to neighboring MlaFB monomers in a similar conformation (*Figure 3B*).

To assess whether similar CTE/'handshake' motifs have been observed in other ABC transporter structures, we used a Dali search (*Holm, 2019*) with our MlaF crystal structure to retrieve all the ABC domains from the PDB and examined the C-terminus of each structure. Most ABC structures have no significant C-terminal region (257 of 379 structures analyzed; see methods). However, 122 ABC structures had an additional small C-terminal domain (CTD), varying in size from about 15–125

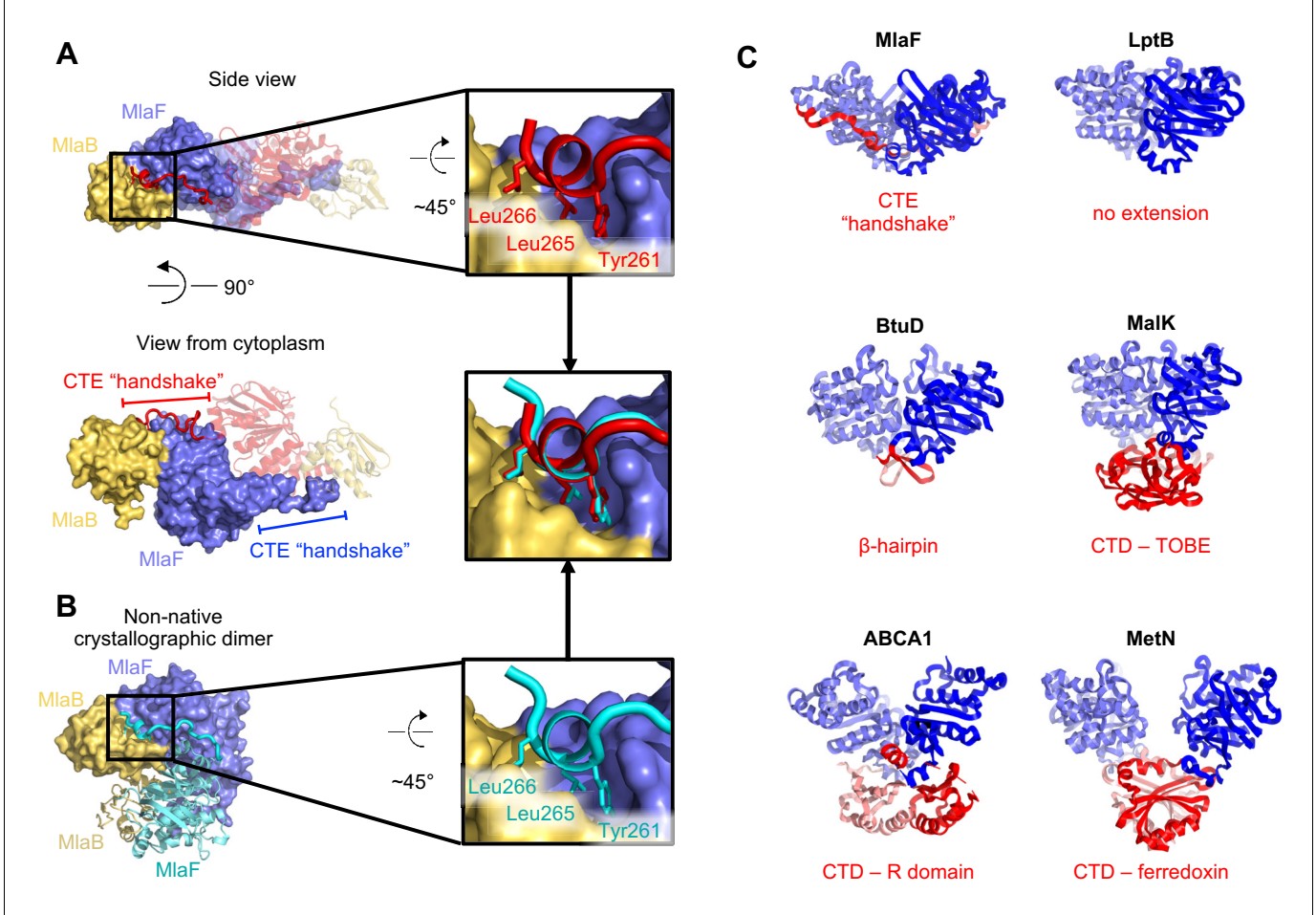

**Figure 3.** C-Terminal Extension (CTE) of MlaF dimer adopts a unique conformation, forming a 'handshake' with the adjacent monomer. (**A**) MlaF$_2$B$_2$ structure with one heterodimer shown in surface representation (MlaF, purple; MlaB, yellow) and the other represented as a cartoon (MlaF, red; MlaB, yellow). The CTE from one MlaF monomer (red) docks at the MlaFB interface of the other MlaF monomer. Inset shows three hydrophobic residues of the CTE that dock at the MlaF/MlaB interface. (**B**) Crystal structure of MlaF$_1$B$_1$ crystallographic (non-native) dimer in which CTE contacts are still maintained. Inset shows the three hydrophobic residues in the CTE, for which interactions to the MlaFB interface are maintained in the non-native arrangement. (**C**) Comparison of C-terminal regions from different ABC transporters, among which the MlaF CTE is unique (PDB IDs: 6MI8 [*Li et al., 2019*], 2QI9 [*Hvorup et al., 2007*], 1Q12 [*Chen et al., 2003*], 5XJY [*Qian et al., 2017*], 3DHW [*Kadaba et al., 2008*]).

residues in length. These domains range from a relatively simple β-hairpin motif in the cobalamin/metal-chelate transporters BtuCDF (*Hvorup et al., 2007*) and HI1470/HI1471 (*Pinkett et al., 2007*), to bonafide domains in MetNI (ferredoxin-like) (*Kadaba et al., 2008*), ABCA1 (R domain) (*Qian et al., 2017*), and MalFGK (*Diederichs et al., 2000*; *Chen et al., 2003*) and ModBC (*Gerber et al., 2008*) (transport-associated oligonucleotide/oligosaccharide binding domain [TOBE]) (*Figure 3C*). The CTDs of other ABC transporters are generally thought to regulate transport activity by promoting or inhibiting the association of the NBDs, either in response to small molecule binding (*Kadaba et al., 2008*; *Gerber et al., 2008*) or interaction with other regulatory proteins (*Chen et al., 2013*; *Ko et al., 2004*). In each of these cases, the CTDs from the complex self-associate just below their associated NBDs, distal from the membrane and TMDs (*Figure 3C*). However, the CTE of MlaF is different in that: 1) it does not form a compact folded domain and 2) the MlaF CTEs do not interact with each other below the transporter complex, as observed for the regulatory CTDs from other transporters, but instead wrap around the sides of the neighboring MlaF and MlaB subunits. Thus, the CTE of MlaF appears to be unique among structurally characterized ABCs.

## MlaF handshake is required for MlaFEDB assembly and function

To test whether the CTE/'handshake' is important for transporter function in cells, we constructed a series of mutations in MlaF and tested their ability to complement an *E. coli* strain with an in-frame chromosomal deletion of the *mlaF* gene (*Figure 4A*). *E. coli* lacking *mlaF* is unable to grow on LB agar in the presence of SDS and EDTA, but this defect can be complemented by providing a wild type copy of *mlaF* on a plasmid (*Figure 4B*). Plasmids encoding catalytically dead MlaF variants, with mutations in either the Walker A or B motif (Lys47Ala and Glu170Gln), failed to complement (*Figure 4B*), consistent with ATP hydrolysis being essential for transport by MlaFEDB (*Thong et al., 2016*). We then tested the impact of point mutations in the three hydrophobic residues in the C-terminal helix, as well as a complete deletion of the CTE (residues 247–269). Single mutations of Tyr261, Leu265, or Leu266 to alanine had no detectable effect on MlaF function in cells and restored growth to wild-type levels (*Figure 4B*). However, an MlaF triple mutant (Tyr261Ala + Leu265Ala + Leu266Ala) only partially restored growth, and MlaF lacking the entire CTE failed to complement the *mlaF* deletion and was therefore completely non-functional (*Figure 4B*). Thus, the CTE is required for MlaF function in *E. coli*, as is ATP binding and hydrolysis.

Since our structure revealed that the CTE of MlaF makes key contacts with MlaB, we hypothesized that the CTE may be required in order to stabilize the MlaF-MlaB interaction. To test this, we co-expressed MlaF and N-terminally His-tagged MlaB in *E. coli* and used a NiNTA/His-MlaB pull-down to detect its interaction with MlaF. WT MlaB pulled down roughly stoichiometric amounts of WT MlaF, as well as MlaF with a mutation in the Walker A or Walker B motifs (*Figure 4C*), indicating that MlaF-MlaB interaction is not dependent upon nucleotide binding/hydrolysis. Surprisingly, none of the mutations in the MlaF CTE, including the complete CTE deletion, had a detectable effect on the MlaF-MlaB interaction (*Figure 4C*). Thus, the essential function of the CTE in cellular assays cannot be explained by its role in stabilizing the MlaF-MlaB interaction.

As part of a larger ABC transporter complex, 2 copies of the MlaF NBD must associate with 2 copies of the MlaE TMD, coupling ATP hydrolysis to lipid transport. To assess whether deletion of the MlaF CTE affects MlaFEDB complex formation or transport activity, we over-expressed the complete MlaFEDB complex (wild type or MlaFΔCTE mutant; *Figure 4D*) and purified the resulting detergent-solubilized complexes via an N-terminal His tag on the MlaD subunit. The wild-type complex yielded the expected banding pattern on SDS-PAGE, with a clear band for each of the four subunits (*Figure 4E*). Strikingly, the sample prepared using the MlaFΔCTE mutant retained clear bands for MlaE and MlaD, but both MlaF and MlaB were absent (*Figure 4E*). Thus, deletion of the MlaF CTE interferes with the ability of the MlaF NBD to stably associate with MlaE and the rest of the Mla-FEDB complex. As MlaB is still capable of binding to MlaFΔCTE (*Figure 4C*), the loss of both proteins from the complex probably reflects their tight association with each other, and the presumed minimal contacts between MlaB and the other transporter subunits (see analysis of MlaF-MlaB interaction below).

In ABC transporters, ATP hydrolysis is used to drive conformational changes in the TMDs that facilitate substrate translocation, and this communication is mediated by the interaction between a cleft in each NBD and a coupling helix on the cytoplasmic side of the TMD (*ter Beek et al., 2014*). Deletion of the MlaF CTE seems unlikely to affect MlaE binding directly, as the location of the CTE in our structure is far away from the predicted site of coupling helix binding (~30 Å away). Alternatively, the reciprocal 'handshake' between the two MlaF subunits in the complex may facilitate NBD dimerization, and thereby promote stable association with the MlaFEDB complex by increasing the valency of the interaction between MlaF and MlaE. If this hypothesis is correct, then artificial dimerization should rescue the function of the MlaFΔCTE mutant. To test this, we fused the GCN4 dimerization domain to the C-terminus of MlaFΔCTE (*Figure 4F*) and assessed the ability of this construct to restore the growth of the *mlaF* knock-out *E. coli* strain in the presence of SDS and EDTA. The GCN4 dimerization domain was connected to MlaFΔCTE either directly, or via a Ser/Gly linker, and linkers of 3 different lengths were tested to ensure that conformational changes in the assembled MlaFEDB transporter were not restricted by the linker length. In contrast to the MlaFΔCTE mutant alone, which did not grow and was indistinguishable from the *mlaF* knock-out strain, the MlaFΔCTE-GCN4 fusion substantially restored growth in the presence of SDS and EDTA (*Figure 4G*), albeit not to the same extent as wild type MlaF. The observation that forced dimerization can largely restore

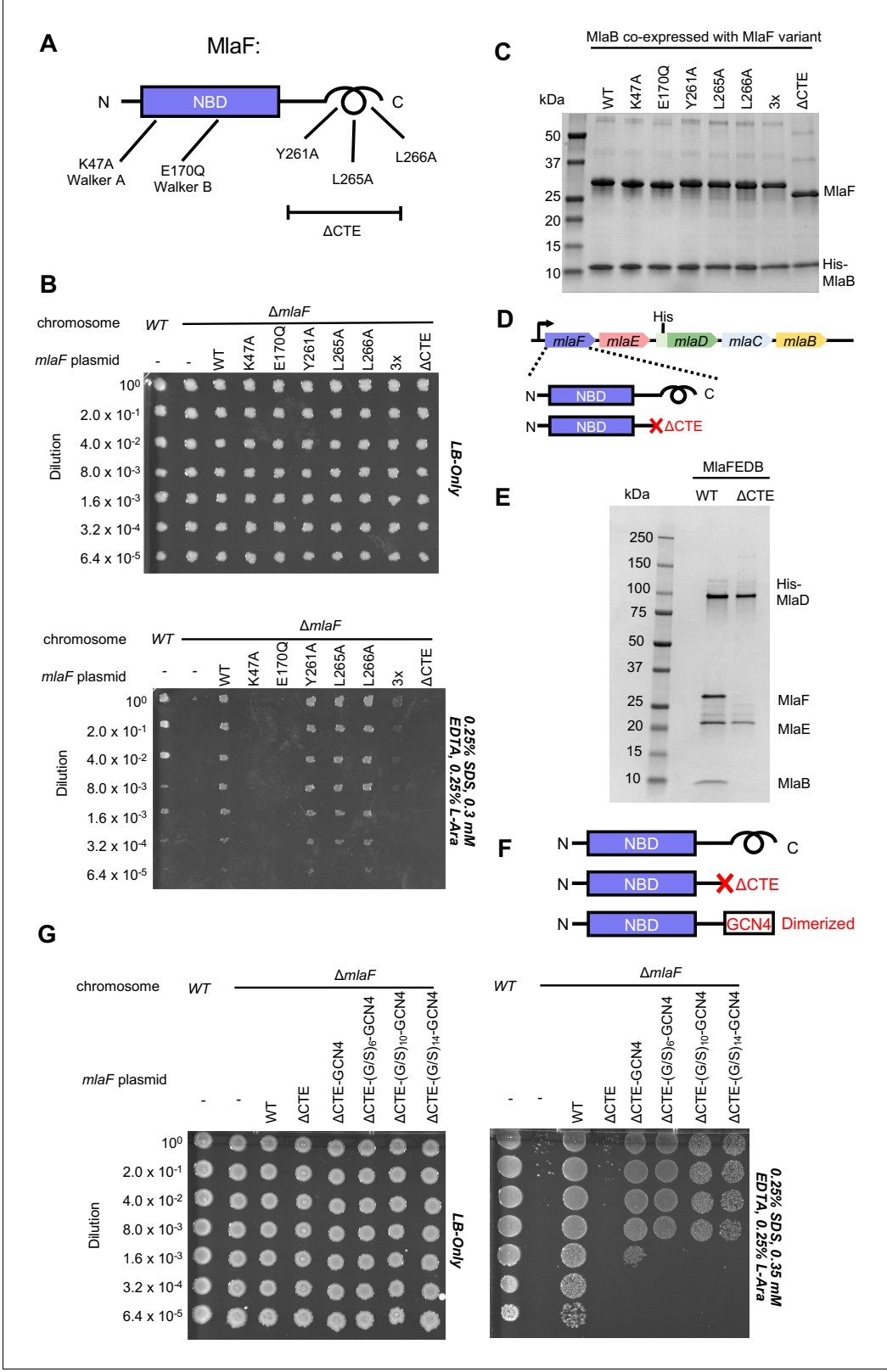

**Figure 4.** Cellular and biochemical assays probing the role of the MlaF CTE. (**A**) Schematic of MlaF depicting wild-type and mutants generated. (**B**) Cellular assay for the function of MlaF. 10-fold serial dilutions of the indicated cultures spotted on LB only (top) or LB plates containing SDS and EDTA at the concentrations indicated (bottom), and incubated overnight. Expression of constructs containing the following mutations fail to rescue: K47A, E170Q,

*Figure 4 continued on next page*

*Figure 4 continued*

3x (Y261A-L265A-L266A triple mutant), ∆CTE. Growth of all strains is unaffected on control plates containing LB only. Results depicted are representative of at least three biological replicates. (C) SDS-PAGE of pull-down experiments showing MlaB and MlaF. WT His-MlaB is the bait, co-expressed with WT or mutant MlaF constructs, as noted in the figure. Results depicted are representative of at least three biological replicates. (D) Schematic of MlaFEDCB operon, depicting WT MlaF and MlaF ∆CTE. (E) SDS-PAGE of protein purified from the constructs depicted in (D). Results depicted are representative of at least three biological replicates. (F) Schematic of MlaF constructs used in (G) GCN4 was used for artificial dimerization, and the GCN4 dimerization domain was connected to the rest of MlaF via a Ser/Gly linker of different lengths. (G) Cellular assay for the function of MlaF. 10-fold serial dilutions of the indicated cultures spotted on LB only (left) or LB plates containing SDS and EDTA at the concentrations indicated (right), and incubated overnight. MlaF ∆CTE mutant fails to grow on SDS-EDTA plates, and can be partially rescued by GCN4 dimerized MlaF constructs. Growth of all strains are unaffected on control plates containing LB only. Results depicted are representative of at least three biological replicates.

function in MlaF∆CTE suggests that the normal function of the CTE may be to stabilize the MlaF dimer via the reciprocal 'handshake' observed in the crystal structure.

## MlaB interaction with MlaF defines possible regulatory site on an ABC ATPase domain

The function of MlaB in lipid trafficking driven by the MlaFEDB complex is not well understood, but our structure of the $MlaF_2B_2$ subcomplex provides some insights. MlaB consists of a single STAS-like domain (S̲ulfate T̲ransporter and A̲nti-S̲igma factor antagonist [*Aravind and Koonin, 2000*]), consisting of a sandwich of 4-stranded beta-sheet on one side packed against 3 α-helices on the other side (*Figure 5A and B*). In most of the STAS domain structures determined to date (*Jurk et al., 2016*; *Marles-Wright et al., 2008*; *Quin et al., 2012*; *Williams et al., 2019*; *Pasqualetto et al., 2010*; *Babu et al., 2010*; *Serrano et al., 2010*; *Etezady-Esfarjani et al., 2006*; *Kumar et al., 2010*; *Kim et al., 2008*; *Sharma et al., 2011*), such as SpoIIAA (*Masuda et al., 2004*; *Seavers et al., 2001*; *Kovacs et al., 1998*), the C-terminus is longer, adding an additional fifth beta-strand to the edge of the sheet and wrapping around to pack a final α-helix against the opposite face of the sheet (*Figure 5E and F*). However, this fifth strand and additional helix are clearly absent in MlaB, as the C-terminus is too short and the final MlaB residue is clearly resolved in our structure. Thus, MlaB appears to be a minimalist member of the STAS domain family, and is perhaps more structurally similar to the smaller STAS domains associated with some SLC26/SulP transporters (*Nocek et al., 2010*; *Geertsma et al., 2015*; *Figure 5C and D*).

The MlaB binding site on MlaF is centered on the helical subdomain of the core NBD in MlaF, distant from the $MlaF_2$ dimer interface and the ATP binding sites (with ~30 Å separating the nearest atoms in MlaB and the bound ADP). However, the binding site on MlaF actually consists of two main parts, with contributions from each of the two MlaF subunits in the dimer (*Figure 5G and H*). The first part of the interface is mediated by the helical subdomain of MlaF (*Figures 2B* and *5J*), which makes extensive contacts with all three helices from MlaB (*Figure 5H*, blue residues). The second part of the interface is formed by the unique CTE/'handshake' originating from the neighboring MlaF subunit in the complex (*Figure 5H*, black residues). As mentioned above, a short helix from this CTE places three hydrophobic residues in a groove between MlaB and the helical subdomain of MlaF (Tyr261, Leu265, Leu266).

Binding to MlaF is mediated entirely by the helical face of MlaB, including helices α1, α2, and α3 and the C-terminal coil (*Figure 5I,J*, and *Figure 5—figure supplement 1A*). α2 lies between α1 and α3 (*Figure 5B*), and consequently is at the center of the interface and makes the most interactions with MlaF. Residue Thr52 on MlaB is particularly noteworthy, as this residue lies close to the center of the MlaF-MlaB interface (*Figure 5I and J*) and is a site of phosphoregulation in other STAS domain proteins (*Yang et al., 1996*; *Gaidenko et al., 1999*). While there is currently no experimental evidence of phosphorylation at Thr52 in MlaB, our structure predicts that phosphorylation of this site would likely cause steric and electrostatic repulsion. Consistent with this hypothesis, a phosphomimetic Thr52Glu mutation substantially reduced the amount of MlaF associated with His-MlaB in our pull-down assay (*Figure 5K*). However, it should also be noted that a Thr52Ala mutation was previously found to be non-functional in *E. coli* and resulted in a greatly reduced ATPase activity of the

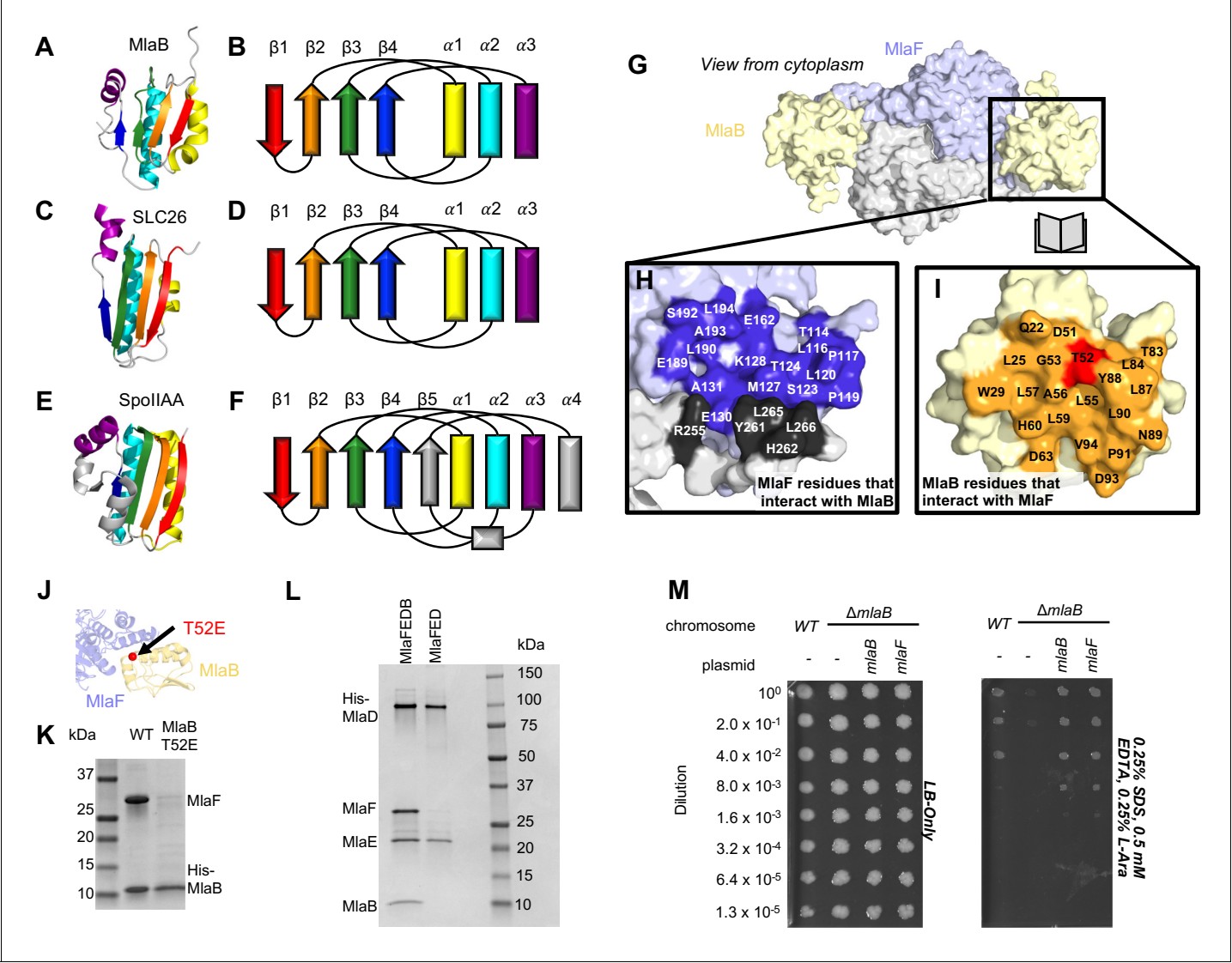

**Figure 5.** Role of MlaB in complex stability. Cartoon representation of MlaB (**A**) and other STAS-domain proteins, SLC26 (**C**, PDB: 5DA0 [*Geertsma et al., 2015*]) and SpoIIAA (**E**, PDB: 1TH8 [*Masuda et al., 2004*]), and corresponding topology diagrams (**B, D, F**, respectively). (**G**) Surface representation of the MlaFB dimer. Insets: (**H**) MlaF surface; one monomer is shown in lilac and the other in light grey. Residues that interact with MlaB to form the interface are shown in dark purple and dark grey for each monomer, respectively (**I**) MlaB surface (pale yellow). Residues that interact with MlaF to form the interface are shown in orange. T52 is shown in red. (**J**) MlaFB is shown in cartoon representation. The position of the T52E mutant on MlaB is shown as a red sphere (**K**) SDS-PAGE of pull-down experiment showing MlaB and MlaF. WT or mutant T52E His-MlaB is the bait, co-expressed with a WT MlaF construct. Results depicted are representative of at least three biological replicates. (**L**) SDS-PAGE of purified protein resulting from MlaFEDCB or MlaFEDC expression constructs in which His-MlaD is used for purification. As expected, MlaC does not co-purify with either construct. MlaF and MlaB are both absent when MlaFEDC is used as the expression construct. Results depicted are representative of at least three biological replicates. (**M**) Cellular assay for the function of MlaB. 10-fold serial dilutions of the indicated cultures spotted on LB only (left) or LB plates containing SDS and EDTA at the concentrations indicated (right), and incubated overnight. MlaB knockout mutant fails to grow on SDS-EDTA plates, and can be rescued by overexpression of either WT MlaB or MlaF. Growth of all strains are unaffected on control plates containing LB only. Results depicted are representative of at least three biological replicates.

The online version of this article includes the following figure supplement(s) for figure 5:

**Figure supplement 1.** Comparison of our *E. coli* MlaFB crystal structure (2.60 Å resolution) and coordinates deposited for *A. baumannii* MlaFB from a cryo EM reconstruction (8.7 Å resolution).

**Figure supplement 2.** Co-expression of MlaB and MlaF tagged at different positions.

MlaFEDB complex, yet MlaB remained stably bound to MlaF (*Thong et al., 2016*). Thus, mutations at this site may alter MlaF function in ways that are difficult to reconcile with simple phosphoregulation.

Our structure differs substantially from the model proposed for the MlaF$_2$B$_2$ subcomplex from *Acinetobacter baumannii* (*Kamischke et al., 2019*). Relative to our crystal structures of the *E. coli* proteins, the MlaB subunit of the *A. baumannii* EM model is rotated by approximately 120˚ (*Figure 5—figure supplement 1A*). As a consequence, the residues on MlaB that would mediate binding to MlaF are completely different in the *E. coli* and *A. baumannii* models, with the inferred contact surfaces lying on opposite sides of MlaB (see methods; *Figure 5—figure supplement 1B*). While the binding mode for *E. coli* MlaF and MlaB is unambiguously defined in our electron density map, down to the rotameric state of most side chains throughout our model (*Figure 5—figure supplement 1B*), the *A. baumannii* model is based upon a much lower resolution map that lacks clear secondary structure features (8.7 Å, EMD-0007), and it is possible that MlaB could not be docked unambiguously. An EM map of the *E. coli* MlaFEDB complex of similar quality was also previously reported (*Ekiert et al., 2017*), but the relative orientation of MlaB could not be determined reliably due to the low resolution. However, we cannot exclude the possibility that the MlaF-MlaB interaction is completely different in *E. coli* vs *A. baumannii*; while the MlaF subunits are fairly well conserved (~59% identical), the MlaB subunits are much more divergent (<20% identical) (*Figure 5—figure supplement 1C and D*).

Our structure shows that MlaF interacts and functions in concert with MlaB, and that this interaction is mediated in part by an elongated C-terminal extension. To assess whether other STAS domain proteins interact with ABC transporters in a similar way, we examined all the entries in the PDB that contain either STAS or ABC domains (entries recovered via Dali search; see methods). First, we searched this dataset for any entries that contained both STAS and ABC domains, but were unable to identify any other examples of STAS and ABC domains in the same PDB entry. Next, we reasoned that unrelated, non-STAS domain proteins may bind to other ABC proteins on the MlaF surface recognized by MlaB and thereby regulate their function in an analogous way. However, we were unable to identify any other ABC structure in the PDB with a binding partner present in the vicinity of the MlaB binding site. Thus, the interaction of MlaB with MlaF appears to define a previously undescribed binding site on ABC transporters, potentially opening up a novel mechanism for the regulation of ABC transporter activity.

## MlaB binding may regulate MlaF stability and function

Binding of the MlaB STAS domain to the MlaF NBD has the potential to regulate the ATPase and transporter activity of the MlaFEDB complex. For example, the helical subdomain of ABC proteins is known to rotate relative to the catalytic domain (*Karpowich et al., 2001*; *Smith et al., 2002*; *Orelle et al., 2010*), which reconfigures the ATPase active sites and also alters the cleft involved in the interaction with the coupling helix of the TMD. The function of STAS domains is best understood in the context of prokaryotic transcriptional regulation, but they also play a less-well understood regulatory role in the function of SulP/SLC26 family of secondary active transporters from bacteria to humans. In *B. subtilis*, the prototypical STAS domain protein, SpoIIAA, binds to and inhibits the activity of its binding partner, an anti-Sigma factor, as part of a complex network that ultimately regulates transcription of genes important for spore development.

Based upon the known mechanism of SpoIIAA, we hypothesized that the MlaB STAS domain may regulate the level of MlaF activity in the cell. To test this hypothesis, we first set out to express and purify transporter complexes for functional studies, including the complete IM complex (MlaFEDB) as well as a complex lacking MlaB (MlaFED). In both cases, the periplasmic protein MlaC was also co-expressed due to its presence in the middle of the operon, but it does not stable interact with other Mla proteins. When MlaF, MlaE, MlaD, and MlaB were co-expressed and purified via a His-tag on MlaD, a stable complex was isolated with the expected stoichiometry of roughly MlaF$_2$E$_2$D$_6$B$_2$ (*Thong et al., 2016*; *Ekiert et al., 2017*). Yet to our surprise, co-expression of MlaF, MlaE, and MlaD in the absence of MlaB yielded a complex containing only MlaE and MlaD after affinity purification via the His-tag on MlaD. MlaF, though co-expressed, did not stably associate with MlaE and MlaD in the absence of MlaB (*Figure 5L*). Similar results were previously reported, when MlaF and MlaD were co-expressed with tagged MlaE (*Thong et al., 2016*). Thus, MlaB appears to be required for association of the MlaFB module with the transmembrane subunits MlaE and MlaD.

MlaB could promote stable association of MlaFB with MlaED in at least two ways. First, MlaB could increase the affinity of MlaFB for MlaED. For example, MlaB could interact directly with MlaE or MlaD, or alter the affinity of the MlaF-MlaED interaction (allosterically, or through stabilizing interactions with the CTE/'handshake'). Second, MlaF may be unstable in isolation and may be degraded or aggregate in the absence of MlaB, thereby failing to associate with MlaED. In order to test these two possibilities, we attempted to express and purify MlaF and MlaB separately for biochemical characterization and binding experiments. MlaB was expressed at high levels and could be purified (*Figure 5—figure supplement 2A*), though it was prone to precipitation when concentrated during the purification process. In contrast, we were unable to express and purify significant amounts of MlaF alone with either an N-terminal or C-terminal His tag. When the same tagged MlaF constructs were co-expressed with MlaB, the MlaFB complex could be purified in good yield (*Figure 5—figure supplement 2B*), indicating that tags on MlaF are well tolerated at these positions. These data suggest that MlaB stabilizes MlaF, thus allowing the assembly of MlaFEDB transporters in *E. coli*.

While MlaB appears to be important for MlaF stabilization and transporter assembly, it may have other critical functions in lipid transport. However, if MlaB is merely required for MlaF stability and has no other essential function in transport driven by Mla, then it should be possible to compensate for the loss of MlaB by increasing the amount of active MlaF in the cell, thus driving complex formation with MlaED. To test this, we transformed the *mlaB* KO *E. coli* strain with a wild type copy of *mlaF* under the control of an arabinose-inducible promoter. Whereas the *mlaB* KO was unable to grow on LB agar in the presence of SDS and EDTA, growth was restored to wild type levels by providing either *mlaB* or *mlaF* in a plasmid (*Figure 5M*). Thus, over-expression of MlaF can fully suppress the defect of the *mlaB* knock-out strain. This suggests that MlaB may have no essential function in Mla lipid transport apart from its role in promoting MlaF stability and/or association with the MlaFEDB complex. Thus, rather than activating or inhibiting the MlaF ATPase activity directly, MlaB may regulate overall transporter activity indirectly, by modulating the amount of MlaF in the cell that is available to interact with MlaED and drive transport.

## Discussion

The Mla pathway may function either to maintain outer membrane homeostasis/asymmetry (retrograde transport) (*Malinverni and Silhavy, 2009*; *Thong et al., 2016*; *Chong et al., 2015*; *Sutterlin et al., 2016*) or to export nascent phospholipids to the outer membrane (anterograde transport) (*Kamischke et al., 2019*; *Hughes et al., 2019*). In both cases, one might predict that higher levels of Mla transport activity will be required during conditions of rapid growth or envelope stress, while transport needs may be minimal when growth is limited. Thus, cells will likely require mechanisms to modulate Mla transport activity based upon the cellular metabolic state. In addition to transcriptional/translational regulation of the MlaFEDCB operon, our study of the MlaFB complex suggests a possible role for MlaB as a regulator of MlaF stability, and consequently, the overall transport activity through the Mla pathway (*Figure 6*). Under conditions requiring elevated lipid flux through the Mla pathway, MlaB may be expressed and available for binding to MlaF, leading to MlaF stabilization and assembly of active MlaFEDB transporter complexes (*Figure 6A*). In contrast, under conditions where little lipid transport is required, reduction in MlaB levels may leave unbound MlaF susceptible to degradation (*Figure 6B*). Depletion of MlaF from cells would reduce the proportion of active MlaFEDB transporter complexes capable of hydrolyzing ATP to drive lipid translocation. Currently, there are no other examples of ABC transporters where the stability and/or association of the NBD with the TMD has been proposed to be a major mechanism of regulation. The interaction between MlaB and MlaF is unlike anything observed previously in structurally characterized ABC transporters, and may represent a largely unexplored mechanism to indirectly regulate the ATPase activity of this diverse transporter family. Indeed, the STAS domain from the SLC26 transporter family has been implicated in the regulation of the ABC transporter/chloride channel CFTR (*Ko et al., 2004*), though the structural basis for this interaction and gating remain unknown.

Precisely how the levels of 'active' MlaB might be regulated is an open question. One particularly intriguing possibility is that MlaB function may be regulated by phosphorylation. At least three other bacterial STAS domain proteins are known to be regulated by Ser/Thr phosphorylation: SpoIIAA (*Duncan and Losick, 1993*; *Min et al., 1993*), RsbR (*Gaidenko et al., 1999*), and RsbS (*Yang et al., 1996*). SpoIIAA is part of a regulatory circuit in *B. subtilis* that controls the sporulation process, while

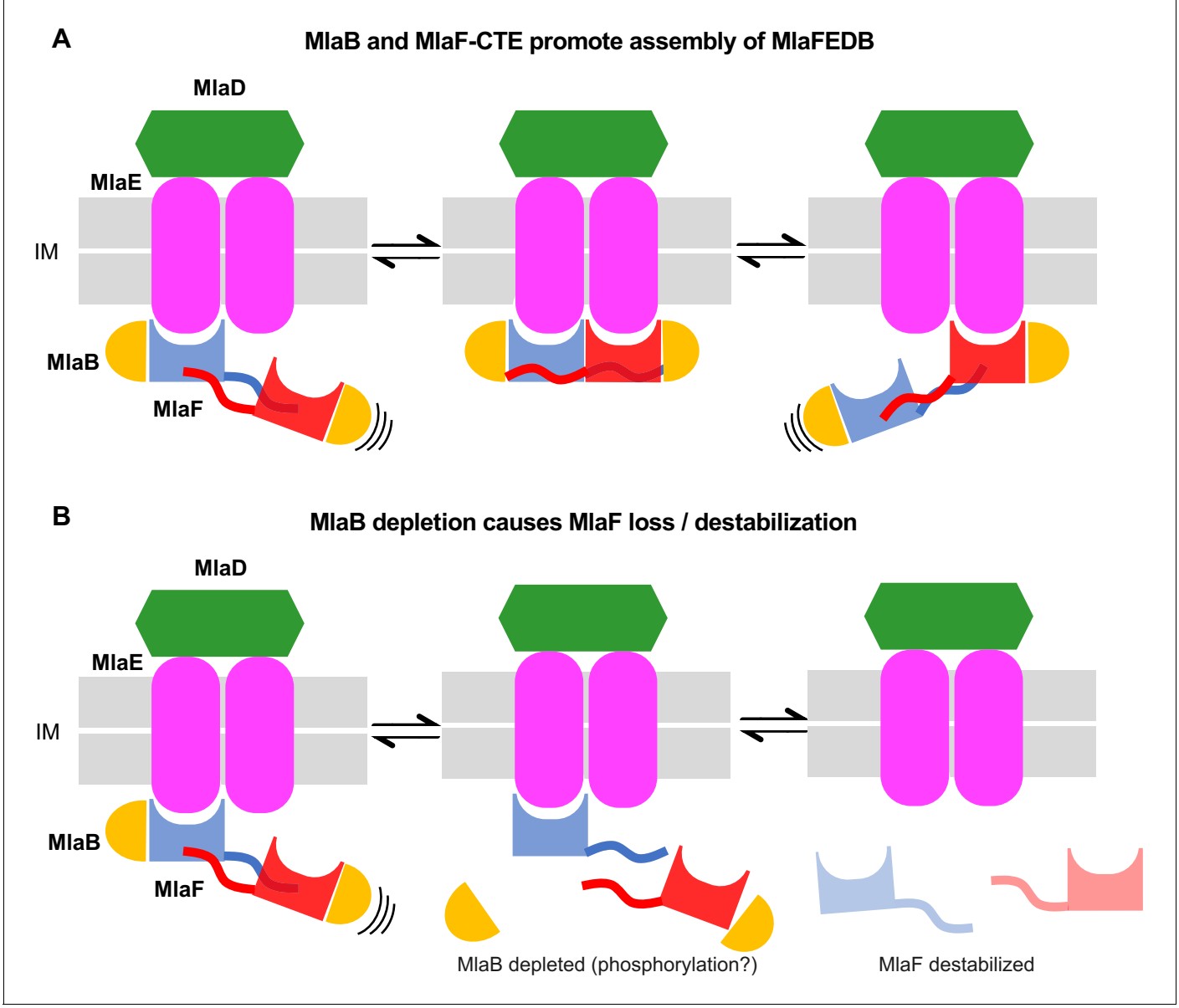

**Figure 6.** Model for the role of MlaFB in the MlaFEDB complex. (**A**) The handshake of the MlaF CTE C-Terminal Extension (CTE) of MlaF stabilizes the MlaF dimer and its association with the MlaFEDB complex. (**B**) The presence of MlaB, and the MlaFB interaction, is required for MlaF association with the MlaFEDB complex. In the absence of MlaB, MlaF does not associate stably with MlaE, and therefore the MlaFEDB complex would be inactive without its ATPase subunit bound.

RbsR and RbsS are components of the bacterial 'stressosome', a ~ 1.8 megadalton cage-like signalling complex (*Marles-Wright et al., 2008*; *Williams et al., 2019*). All three of these proteins are phosphorylated on a conserved, structurally equivalent Ser or Thr residue, and phosphorylation disrupts binding to their interaction partners. A Ser/Thr residue is broadly conserved among STAS domain proteins, including MlaB (Thr52), suggesting that phosphoregulation may be a common theme among diverse STAS domain proteins. However, it remains to be determined whether MlaB can be phosphorylated to a significant degree in cells, and future studies will be necessary to define if and how this potential phosphoregulation may occur. Alternatively, levels of active MlaB may be regulated by several other possible mechanisms at the level of transcription, translation, or protein degradation. Some STAS domains have also been proposed to bind ions (*Lolli et al., 2016*), nucleotides (*Sharma et al., 2011*; *Buttani et al., 2006*; *Avila-Pérez et al., 2009*), or fatty acids

(*Babu et al., 2010*), leaving open the possibility that MlaB may respond to small metabolites or even transporter substrates and act as a switch to turn transport on or off.

Our data indicate that the CTE/'handshake' of MlaF plays a critical role, as truncation of the CTE completely abolishes Mla function despite having no apparent effect on protein folding or the MlaF-MlaB interaction itself. As artificial dimerization of the truncated MlaF can restore MlaF function to a large degree, we propose that the reciprocal handshake of the CTEs from adjacent MlaF subunits promotes NBD dimerization, which in turn increases the effective affinity of MlaF for the MlaE (*Figure 6*). It is unclear why such a mechanism may have evolved, as all other ABC transporter structures appear to lack such a handshake motif. However, the dimeric regulatory domains of some transporters, such as MetNI, have been proposed to regulate the configuration of the NBDs (*Kadaba et al., 2008*), and perhaps might even modulate the monomer-dimer equilibrium in response to ligands. By analogy, changes in the levels of active MlaB in the cell may similarly regulate the propensity of MlaF to dimerize, as MlaB interacts with several key residues in the CTE. In the absence of MlaB, the handshake between MlaF subunits would likely be disrupted, disfavoring the dimer and facilitating the release of monomeric MlaF from the transporter complex, perhaps facilitating MlaF turn-over. Further studies will be required to separate the relative contributions of MlaB and the CTE, and how they may synergize to tune the activity of the Mla transport system in the cell.

## Materials and methods

### Bacterial culture and molecular biology

*mlaB* and *mlaF* derivatives of *E. coli* strains MG1655 (*Blattner et al., 1997*) and BW25113 (*Datsenko and Wanner, 2000*) were constructed by P1 transduction from strain JW5535 (Δ*mlaB::gfp*,CAT) (*Mori et al., 2015*) or the Keio strain JW3162 (Δ*mlaF*:kan) (*Baba et al., 2006*) followed by excision of the antibiotic resistant cassettes using pCP20 (*Cherepanov and Wackernagel, 1995*). The *mlaF* allele is an in-frame deletion with an FRT scar, while the *mlaB* allele is an in-frame replacement of the *mlaB* gene by *gfp*. All plasmids and plasmid-borne mutations were constructed using Gibson assembly (*Gibson et al., 2009*).

SDS+EDTA sensitivity was assayed in LB agar supplemented with 0.5% SDS and 0.3–0.4 mM EDTA for *mlaF* strains and 0.5% SDS and 0.4–0.5 mM EDTA for *mlaB* strains. The final concentration of EDTA used here is significantly lower than the concentrations reported in other studies of *mla* mutants (*Malinverni and Silhavy, 2009*; *Thong et al., 2016*), including experiments conducted by us with the same strains at a different institution with different reagents (*Ekiert et al., 2017*). We find that this growth assay is very sensitive to the reagents used, particularly the LB agar (see *Figure 1—figure supplement 1*. For the experiments reported here, we used Difco LB agar pre-mix (BD Difco #244510), a 10% stock solution of SDS (Sigma L5750), and a 500 mM stock solution of EDTA, pH 8.0 (Sigma ED2SS). Several sources and purity of SDS yielded comparable results. For complementation experiments, *E. coli mlaF* or *mlaB* strains were transformed with a pBAD-derived plasmid (Invitrogen) encoding WT *mlaF* or *mlaB* genes, or the indicated mutations. Serial dilutions of the resulting strains in 96 well plates were replicated using a 96 pin manifold/replicator, or manually spotted (2 uL each), on plates containing LB agar or LB agar supplemented with 0.5% SDS and 0.25–0.50 EDTA, and incubated for 16–24 hr.

### Expression and purification of MlaFB

DNAs corresponding to the full-length MlaF (residues 1–269) and N-terminally His-TEV-tagged MlaB (MHHHHHHENLYFQ followed by MlaB residues 2–97) were amplified from pBEL1195 and cloned by Gibson assembly into a custom pET vector (pDCE467) to create a bicistronic expression cassette. The resulting plasmid (pBEL1307) was transformed into Rosetta 2 (DE3) cells (Novagen). For expression, an overnight culture of Rosetta 2 (DE3)/pBE1307 was grown at 37° C with shaking to an OD600 of ~0.9, then induced by addition of IPTG to a final concentration of 1 mM and continued incubation for 4 hr shaking at 37° C. Cultures were harvested by centrifugation, and pellets were resuspended in lysis buffer (50 mM Tris pH 8.0, 300 mM NaCl, 10 mM imidazole). Cells were lysed by two passes through an Emulsiflex-C3 cell disruptor (Avestin), then centrifuged at 38,000 g to pellet cell debris. The clarified lysates were loaded onto a Qiagen Superflow NiNTA cartridge on a Bio-Rad NGC FPLC system, washed with Ni Wash Buffer (50 mM Tris pH 8.0, 300 mM NaCl, 10 mM imidazole),

and bound proteins eluted with Ni Elution Buffer (50 mM Tris pH 8.0, 300 mM NaCl, 250 mM imidazole). MlaFB containing fractions eluted from the NiNTA column were pooled and concentrated before separation on a Superdex 200 16/60 gel filtration column (GE Healthcare) equilibrated in 20 mM Tris pH 8.0, 150 mM NaCl, 2 mM TCEP.

For testing the impact of tag placement on the expression of MlaFB, we created similar expression constructs to pBEL1307, but with His tags on the C-terminus of MlaB (pBEL1308), or the N- or C-terminus of MlaF (pBEL1305 and pBEL1306). These plasmids were transformed into Rosetta 2 (DE3) cells (Novagen) and were expressed and purified by Ni affinity chromatography as described above for pBEL1307.

## Crystallization and structure determination of MlaFB

Gel filtration fractions containing purified MlaFB complex were concentrated to 24 mg/mL. The concentrated material was used to prepare two samples: 1) MlaFB-apo (no additions), and 2) MlaFB +ADP (ADP and magnesium acetate each added to a final concentration of 2 mM). Sitting drop, vapor-diffusion crystallization trials were conducted using the JCSG Core I-IV screens (Qiagen), setup using a Mosquito Crystal (TTP) and incubated/imaged using Rock Imager 1000 systems at 4° C and 18° C. Crystals of MlaFB-apo grew at 18° C from drops consisting of 100 nL protein plus 100 nL of a reservoir solution consisting of 0.7 M ammonium dihydrogen phosphate, 0.07 M sodium citrate pH 5.6, and 30% glycerol, and were cryoprotected by supplementing the reservoir solution with 5% ethylene glycol. Crystals of MlaFB+ADP grew at 4° C from drops consisting of 100 nL protein plus 100 nL of a reservoir solution consisting of 0.2 M magnesium formate, and were cryoprotected by supplementing the reservoir solution with 35% ethylene glycol. Native diffraction dataset were collected at ALS beamline 8.3.1, and indexed to $P2_12_12_1$ (MlaFB-apo) and $P3_221$ (MlaFB+ADP) and reduced using XDS (RRID:SCR_015652) (*Supplementary file 3*; *Kabsch, 2010*).

Initial attempts to phase either dataset by molecular replacement using Phaser (*McCoy et al., 2007*) (RRID:SCR_014219) were unsuccessful, despite numerous attempts and search model preparations. Fortunately, the MlaFB+ADP dataset was successfully solved using BALBES (*Long et al., 2008*) (RRID:SCR_018763), which identified 2OUK (*Schneider et al., 2012*) and 3F43 as successful search models for MlaF and MlaB, respectively. The 3F43 structure of a STAS domain protein from *Thermotoga maritima* was determined at the Joint Center for Structural Genomics (JCSG) without an associated publication. The resulting MlaFB model was adjusted in Coot (*Emsley et al., 2010*) and refined using phenix.refine (*Adams et al., 2010*) (RRID:SCR_016736). The final MlaFB+ADP (MlaF$_2$B$_2$, PDB code: 6XGY) model consists of a single copy each of MlaF and MlaB in the asymmetric unit. Application of the 2-fold symmetry operator results in the biologically relevant, dimeric ABC complex. The refined MlaFB+ADP model was subsequently used to phase the MlaFD-apo dataset using Phaser (*McCoy et al., 2007*), followed by adjustment in Coot (*Emsley et al., 2010*) (RRID: SCR_014222) and refinement using phenix.refine (*Adams et al., 2010*). The final MlaFB-apo (MlaF$_1$B$_1$, PDB code: 6XGZ) model consists of 4 copies of MlaF and 4 copies of MlaB in the asymmetric unit. The resulting coordinates for MlaFB+ADP and MlaFB-apo were validated using the JCSG QC Server v3.1 (https://smb.slac.stanford.edu/jcsg/QC) (RRID:SCR_008251) and Molprobity (*Williams et al., 2018*) (RRID:SCR_014226).

## Bioinformatics (alignments and analysis; structure search and comparisons; DALI)

Structure figures were prepared with PyMOL (Schrödinger) (RRID:SCR_000305), which was also used for structure visualization and analysis, along with Chimera (*Pettersen et al., 2004*). To compare MlaF and MlaB to previously determined structures in the PDB, Dali searches (*Holm, 2019*) (RRID: SCR_013433) of the full PDB initiated with either MlaB or MlaF on Aug. 14, 2019. Hits were manually inspected to confirm they represented bonafide MlaF/ABC or MlaB/STAS homologs, and the curated list of hits was used for subsequent structural and sequence analysis. ABC domains were considered to have a CTD/CTE if their C-terminus extended more than ~15 residues beyond the pair of short helices that mark the end of the ABC domain (i.e., beyond ~Asp245 in MlaF). Sequence alignments were carried out using MUSCLE (*Edgar, 2004*) (RRID:SCR_011812) or Clustal Omega (*Sievers et al., 2011*) (RRID:SCR_001591), and visualized using JalView (*Waterhouse et al., 2009*) (RRID:SCR_006459).

As the deposited coordinates for the *A. baumannii* MlaFEDB complex were modelled with a poly-glycine backbone, standard methods for analyzing protein-protein interactions did not provide meaningful estimates of the contact surfaces between MlaF and MlaB due to the lack of side chains. For example, using a 4 Å cutoff distance, only three atoms in MlaB are found to interact with MlaF. To estimate a plausible contact surface between MlaF and MlaB from these poly-glycine coordinates, we increased the threshold used to define the maximum distance between contacting residues to 8 Å to account for missing side chains. This threshold resulted in a patch of ~20 residues on MlaB sandwiched at the interface with MlaF, and includes the MlaB residues most likely to be making contacts with MlaF. These putative contact residues based upon the *A. baumannii* MlaFEDB model are displayed in *Figure 5—figure supplement 1*.

## MlaFB Pull-Down assay

A derivative of pBEL1307 that co-expresses His-TEV-MlaB and Strep-MlaF was cloned by PCR and Gibson assembly, yielding pBEL1957. Mutations of interest in MlaF or MlaB were subsequently introduced into pBEL1957 via PCR and Gibson assembly. Overnight cultures of Rosetta 2 (DE3) carrying pBE1957 or the desired mutants were diluted ~1:33 in 10 mL LB and grown at 37°C with shaking to an OD600 of ~0.8, then induced by addition of IPTG to a final concentration of 1 mM and continued incubation for 4 hr shaking at 37°C. Cultures were harvested by centrifugation, and pellets were resuspended in 1 mL of freeze-thaw lysis buffer (50 mM Tris pH 8.0, 300 mM NaCl, 10 mM imidazole, 1 mg/ml lysozyme, 0.5 mM EDTA, 25U benzonase) and incubated on ice for 1 hr prior to lysing. Cells then underwent eight freeze-thaw cycles by alternating between liquid nitrogen and a 37°C heat block, then centrifuged at 15,000 g to pellet cell debris. The clarified lysates were incubated with Ni-NTA resin (GE Healthcare) at 4°C, which was subsequently washed with Ni Wash Buffer (50 mM Tris pH 8.0, 300 mM NaCl, 10 mM imidazole) and bound proteins eluted with Ni Elution Buffer (50 mM Tris pH 8.0, 300 mM NaCl, 250 mM imidazole). Elution fractions containing MlaFB were collected and boiled before separation on SDS-PAGE and visualization by colloidal Coomassie staining with InstantBlue (Expedeon).

## Purification of MlaFEDB complexes (WT and mutants)

Plasmid pBEL1200 (*Ekiert et al., 2017*), which encodes the entire *mlaFEDCB* operon with an N-terminal His-TEV tag on MlaD, was modified to remove the CTE of MlaF (pBEL1514) or the mlaB ORF (pBEL1244) by PCR and Gibson assembly (*Gibson et al., 2009*). Overnight cultures of Rosetta 2 (DE3)/pBE1200, Rosetta 2 (DE3)/pBEL1514, and Rosetta 2 (DE3)/pBE1244 were diluted 1:100 into 12 L of LB+Carb+Cm and grown at 37°C with shaking to an OD600 of ~0.8, then induced by addition of Arabinose to a final concentration of 0.2% and continued incubation for 4 hr shaking at 37°C. Cultures were harvested by centrifugation, and pellets were resuspended in lysis buffer (50 mM Tris pH 8.0, 300 mM NaCl, 10% glycerol). Cells were lysed by two passes through an Emulsiflex-C3 cell disruptor (Avestin), then centrifuged at 15,000 g at 4°C to pellet cell debris. The clarified lysates were transferred to ultracentrifuge tubes and centrifuged at 200,000 g at 4°C to pellet the membrane fraction. Membrane pellets were resuspended in membrane resuspension buffer (66.7 mM Tris pH 8.0, 400 mM NaCl, 13.3% glycerol), then n-dodecyl -D-maltopyranoside (DDM; Anatrace) was added to a final concentration of 25 mM. After an overnight incubation with rocking at 4°C, the DDM solubilized membrane fractions were centrifuged at 200,000 g at 4°C. The supernatants were loaded onto a gravity column packed with with Ni-NTA resin (GE Healthcare), which was subsequently washed with membrane Ni Wash Buffer (50 mM Tris pH 8.0, 300 mM NaCl, 40 mM imidazole, 10% glycerol, 0.5 mM DDM) and bound proteins eluted with membrane Ni Elution Buffer (50 mM Tris pH 8.0, 300 mM NaCl, 250 mM imidazole, 10% glycerol, 0.5 mM DDM). Eluates from the NiNTA column were concentrated before separation on a Superdex 200 gel filtration column (GE Healthcare) equilibrated in 20 mM Tris pH 8.0, 150 mM NaCl, 10% glycerol, 0.5 mM DDM. Peak fractions from gel filtration were not boiled before separation on an SDS-PAGE gel and visualization by colloidal Coomassie staining with InstantBlue (Expedeon).

## Individual expression of MlaB and MlaF

Overnight cultures of Rosetta 2 (DE3)/pBE1840 or Rosetta 2 (DE3)/pBEL1246 were grown at 37°C with shaking to an OD600 of ~0.9, then induced by addition of IPTG to a final concentration of 1

mM and continued incubation for 4 hr shaking at 37°C. Cultures were harvested by centrifugation, and pellets were resuspended in lysis buffer (20 mM Tris pH 8.0, 300 mM NaCl, 10% glycerol). Cells were lysed by two passes through an Emulsiflex-C3 cell disruptor (Avestin), then centrifuged at 15,000 g at 4°C to pellet cell debris. The clarified lysates were incubated with Ni-NTA resin (GE Healthcare) at 4°C, which was subsequently washed with Ni Wash Buffer (50 mM Tris pH 8.0, 300 mM NaCl, 10 mM imidazole) and bound proteins eluted with Ni Elution Buffer (50 mM Tris pH 8.0, 300 mM NaCl, 250 mM imidazole). Elution fractions containing MlaF or MlaB were collected and kept on ice while assays were run. Samples were boiled for SDS-PAGE.

## Acknowledgements

We thank C Waddling for support with crystallization; J Holton and G Meigs of ALS 8.3.1 for beamline support; A Gray and C Gross (UCSF) for bacterial strains; and R Redler, J Ilmain, D Puvanendran, and C Vieni for critical reading of the manuscript. This work was supported by NIH grant R35GM128777 (DCE) and NIH grant R00GM112982 and Damon Runyon Cancer Research Foundation grant DFS-20-16 (GB). GLI is supported by an American Heart Association postdoctoral fellowship (20POST35210202). MRM is supported by an NIH T32 predoctoral training grant (T32 GM088118). Beamline 8.3.1 at the Advanced Light Source is operated by the University of California Office of the President, Multicampus Research Programs and Initiatives grant MR-15–328599, the National Institutes of Health (R01 GM124149 and P30 GM124169), Plexxikon Inc, and the Integrated Diffraction Analysis Technologies program of the US Department of Energy Office of Biological and Environmental Research. The Advanced Light Source (Berkeley, CA) is a national user facility operated by Lawrence Berkeley National Laboratory on behalf of the US Department of Energy under contract number DE-AC02-05CH11231, Office of Basic Energy Sciences.

## Additional information

### Funding

| Funder | Grant reference number | Author |
| --- | --- | --- |
| American Heart Association | 20POST35210202 | Georgia L Isom |
| National Institutes of Health | T32 GM088118 | Mark R MacRae |
| National Institutes of Health | R35GM128777 | Damian C Ekiert |
| Damon Runyon Cancer Research Foundation | DFS-20-16 | Gira Bhabha |
| National Institutes of Health | R00GM112982 | Gira Bhabha |

The funders had no role in study design, data collection and interpretation, or the decision to submit the work for publication.

### Author contributions

Ljuvica R Kolich, Ya-Ting Chang, Formal analysis, Validation, Investigation, Writing - review and editing; Nicolas Coudray, Formal analysis, Validation, Investigation, Visualization, Writing - review and editing; Sabrina I Giacometti, Mark R MacRae, Evelyn M Teran, Validation, Investigation, Writing - review and editing; Georgia L Isom, Funding acquisition, Validation, Investigation, Writing - review and editing; Gira Bhabha, Conceptualization, Supervision, Funding acquisition, Validation, Investigation, Writing - original draft, Writing - review and editing; Damian C Ekiert, Conceptualization, Formal analysis, Supervision, Funding acquisition, Validation, Investigation, Writing - original draft, Writing - review and editing

### Author ORCIDs

Ljuvica R Kolich ORCID http://orcid.org/0000-0002-6696-9645
Ya-Ting Chang ORCID https://orcid.org/0000-0003-2580-4622
Nicolas Coudray ORCID https://orcid.org/0000-0002-6050-2219

Mark R MacRae  http://orcid.org/0000-0003-4941-9526
Gira Bhabha  http://orcid.org/0000-0003-0624-6178
Damian C Ekiert  https://orcid.org/0000-0002-2570-0404

Decision letter and Author response
Decision letter https://doi.org/10.7554/eLife.60030.sa1
Author response https://doi.org/10.7554/eLife.60030.sa2

## Additional files

### Supplementary files

- Supplementary file 1. Bacterial strains table.
- Supplementary file 2. Plasmids table.
- Supplementary file 3. Data collection and refinement statistics for crystal structures.
- Transparent reporting form

### Data availability

The structure factors and coordinates for crystal structures were deposited in the Protein Data Bank with accession codes 6XGY (dimeric MlaFB with ADP+Mg) and 6XGZ (monomeric MlaFB in apo state). Plasmids generated in this study have been deposited in Addgene. All data generated or analysed during this study are included in the manuscript and supporting files.

The following datasets were generated:

| Author(s) | Year | Dataset title | Dataset URL | Database and Identifier |
|---|---|---|---|---|
| Chang Y, Bhabha G, Ekiert DC | 2020 | Crystal structure of *E. coli* MlaFB ABC transport subunits in the dimeric state | https://www.rcsb.org/structure/6XGY | RCSB Protein Data Bank, 6XGY |
| Chang Y, Bhabha G, Ekiert DC | 2020 | Crystal structure of *E. coli* MlaFB ABC transport subunits in the monomeric state | https://www.rcsb.org/structure/6XGZ | RCSB Protein Data Bank, 6XGZ |

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
