## [Decision Letter]

[Editors' note: this paper was reviewed by Review Commons.]

**Acceptance summary:**

The paper is a high-quality structure-function study of an ABC transporter delivering lipids to the outer membrane of bacteria and, as such, a potential target for antibiotics. The current study is a continuation of an earlier work, where the authors reported a structure of the entire transport complex (MlaF_2_E_2_D_6_B_2_). However, the cytosolic subunits, comprising pairs of nucleotide binding subunits (MlaF) and regulatory subunits (MlaB), were not well defined. The F_2_B_2_ structure reveals unique structural features that have not been observed for any ABC proteins so far. These include a ‘handshake’ interaction of a C-terminal extension of MlaF and the interactions with a STAS domain protein MlaB. The importance of the latter interaction is that several ABC transporters have been shown to interact with STAS domain proteins in a functionally relevant manner.

---

## [Author Response]

Reviewer #1 (Evidence, reproducibility and clarity):This is a very nice paper giving insight into the details of part of a multi-domain machine that is responsible for sustaining the integrity the outer membrane in Gram-negative bacteria. The focus is on regulatory interactions between two components – MlaF, the nucleotide binding domain of the transporter, and MlaB, a peripherally associated partner. The role of MlaB was not so well understood previously, but this study provides new insight into it regulatory roles through pre-organisation of MlaB, and opens the possibility for covalent modification as a possible controlling process. The data make a compelling case that both the C-terminal tail of MlaF and the interaction with MlaB are required for complete assembly of the MlaFEDB complex and cellular activity.There are a number of minor suggestions:Abstract "and is required for its stability." – maybe changed wording so that it doesn't indicate that it's impacting on protein turnover or denaturation, for example, because these processes were not explored in depth in this report.

We agree. Our experiments show that in the absence of MlaB we can no longer purify significant amounts of MlaF, but it is unclear if this reflects reduced folding, stability, turnover, etc. We have altered the wording in the Abstract to reflect this uncertainty.

The MlaF interface seems to have least buried surface area from visual inspection of Figure 3C, so its dimerisation is critically dependent on the CTE handshake interaction. The CTE handshake is fascinating, and involves interactions with MlaB adjacent to the contacting MlaF.This mode of interaction opens the possibility that MlaB could also help to boost dimerisation of MlaF in the presence of ATP or ADP. Could this be tested? Does ATP or ADP impact on efficiency of pull downs?

We suspect that the reviewer is probably correct. Unfortunately we have been unable to test some of these ideas because we are unable to express and purify MlaF in the absence of MlaB, since MlaB seems to modulate the expression/stability of MlaF. Consequently, it has been difficult to directly assess whether MlaB promotes MlaF dimerization, or increases/decreases the ATPase activity of MlaF (or the MlaFED complex compared to MlaFEDB). In the pull downs, we observe stoichiometric amounts of MlaB and MlaF in the absence of nucleotide, so it would be difficult to see any increase in the interaction in the presence of nucleotide.

Figure 4B: Y261A and L226A seem to work even better than wild type for SDS and EDTA stress response. Any comment on this?

We apologize, it may not have been clear in the PDF, but there are two colonies in the final spot for the WT, and ~2-4 for the Y261A and L226A mutants. We checked all of the replicates of this experiment (2 additional exact replicates plus several additional experiments with slightly different concentrations of SDS+EDTA). None of the mutants grew consistently better than the WT.

The authors make the point that T52 at the MlaB/F interface could be modified for example by phosphorylation as a rapid regulatory signal. Could His60 also be a target – maybe in some two-component signalling event?

Phosphorylation in response to two-component signalling is a very exciting possibility, e.g., in response to outer membrane stress. We focused on T52 because this position is conserved Ser/Thr very broadly among STAS domains and is clearly phosphorylated in other systems, but there are other possible sites, including His60. Looking at 11 MlaB homologs from diverse bacterial species, His60 is conserved in 4/11 sequences (vs. S52/T52 in 11/11 sequences). While that doesn’t mean that His60 is not phosphorylated in *E.* *coli*, it may be less likely, or would be a more species-specific regulatory site.

Query about the x-ray analysis: was a twin fraction of the ADP bound form evaluated? The trigonal space group is susceptible to twinning, which could affect averaging of the tails for instance.

We usually check for twinning immediately after data reduction, but we went back and checked again to be sure. Tests on the ADP data set indicate twinning is unlikely, with an estimated twin fraction of only 2-3%. To gain a more accurate estimate of the twin fraction (if the data set was indeed twinned), we refined our final model against the data set with the twin law applied. This yielded an estimated twin fraction of 3%, with no improvement in Rwork or Rfree. Thus, we conclude that the data set is not twinned, or twinning is very minor.

Reviewer #1 (Significance):The paper will appeal to investigators studying transmembrane transport, and also to wider readership interested in general principles of control and regulation.Reviewer #2 (Evidence, reproducibility and clarity):Ljuvica Kolich et al. and Damian Ekiert present crystal structures of the MlaFB complex, which is a key subcomplex of the MlaFEDB lipid transport complex that maintains lipid distributions between the inner and outer membrane. MalFB is a soluble complex of the ABC family (MalF), with the small MalB subunit being a minimal member of the regulatory STAS protein family. They reveal important features of the quaternary arrangement – an extended C-terminal end of MlaF forms an intermolecular handshake dimerisation module, and they show clear functional effects of mutated, deleted, and dimerisation-substituted constructs. Furthermore, MlaB interacts with a large surface and stabilises the MlaF complex – discussed as an important new regulatory feature in ABC transporters that may also apply to other proteins such as CFTR. All in all the paper presents interesting, novel findings and models. The data on the Thr52 interaction site and target site of phosphorylation-mediated regulation are a bit incomplete:1) Is it possible to show dimerisation directly from purified samples using SEC? the authors describe a monomeric apostructure and a dimeric ADP structure – how is that going on? Why is ADP having such a profound effect on dimerisation? Rotation of the helical subdomain by ~10 degrees is described as a link between subdomain and nucleotide binding – perhaps it could be explained a bit clearer why the apo-state adopts a 1:1 complex.

This is an interesting question, as binding of ATP to ABC ATPases usually leads to dimerization, and following hydrolysis the dimer dissociates to release ADP+Pi, yielding two monomers. What we failed to make clear in our manuscript is that nucleotides didn’t seem to have a major influence on whether we trapped monomer vs. dimer in the crystals. We actually determined structures for MlaFB with ADP in the both the “monomeric” and “dimeric” states, and determined additional structures of “monomeric” MlaFB in the apo and AMPPNP bound states. All of these were co-crystallized, not soaked. Because this suggested that we were not seeing nucleotide-induced changes in the structure, we decided to focus on the highest resolution structures of these two different oligomeric states and do not present the other structures. We would be happy to include these additional structures (all ~3 Å resolution), though we don’t think they would add anything to the manuscript. To make this clearer, we now refer to the MlaF_1_B_1_ (“monomer”) and MlaF_2_B_2_ (“dimer”) structures throughout the text, rather than whether they are apo or ADP bound. On a final note, we have thus far only observed monomeric MlaFB by SEC, even when injected at very high concentrations (~50 mg/mL), so dimerization is probably occurring during crystal growth in this case, and any dimer in solution is probably in rapid equilibrium with monomer. It is also likely that dimerization is stabilized in the context of the entire MlaFEDB complex, when MlaF binds to MlaE.

2) Related to this – is it possible to probe the dimerisation by SEC analysis. Will a sample run in ADP show a F_2_B_2_ dimer peak of the MlaFB complex? Can this be used to verify the effect of mutations, deletions and new designs of the protein constructs? It would seem like a simple control of the dimer-disruption model shown by the growth complementation.

See response to comment 1 above. We expect that the dimer observed in some ADP-bound crystals does not reflect an ability of ADP to promote dimerization. By analogy to other ABC transporters, we expect that ATP binding may facilitate interactions between MlaF subunits, as is typical. We have not detected any MlaF_2_B_2_ dimer peak by SEC, which may reflect the low affinity of between the two MlaF subunits when not bound to the transmembrane subunit, MlaE.

3) The Thr52Glu and Thr52Ala mutations of MlaB are used to confirm the key position of this residue at the MlaFB interface and leads to a speculation of phosphoregulation, which is also observed for other STAS proteins with an equivalent Thr/Ser site. First of all, given the central location in the MlaB fold and at the MlaF interface it cannot surprise that the Thr52Glu and Thr52Ala mutations inhibit the complex formation, but it cannot support the model of phosphoregulation. Perhaps a less disruptive mutation could be introduced, e.g. Thr52Val or -Cys. I would suggest to play down this discussion in the absence of e.g. phosphoproteomics data to qualify the identification of Thr52 as a phosphorylation site in *E. coli*. However, public phosphoproteomics data repositories might even have it. Also identifying th *E. coli* kinase would be very informative.

The reviewer makes several good points here, and we agree that the phosphorylation of MlaB is speculative. The fact that mutation of Thr52 affects function is entirely consistent with its position near the center of the binding interface, and doesn’t shed any particular light on hypothetical phosphorylation. We have made this clear in the Results section, and have used the previously published results from the Thr52Ala mutation to emphasize that something more complicated may be going on here. Speculation regarding phospho-regulation is now confined to the Discussion section. As the reviewer points out, nailing this down will require experimental determination of whether and when Thr52 is phosphorylated, and identification of the kinase. This is an exciting study in its own right, and one that we are pursuing.

Reviewer #2 (Significance):The same lab has released a preprint of the entire MlaFEDB complex determined by cryo-EM at about 3 Å resolution. https://www.biorxiv.org/content/10.1101/2020.06.02.129247v 1  – it would seem important that these reports refer to each other.The dimerisation motif of MlaF and the regulatory, stabilising interaction of MlaB with MlaF are interesting and novel findings.